# Exploring the Life Experiences of Living with Cardiac Arrhythmia Developed During Pregnancy

**DOI:** 10.3390/healthcare12212178

**Published:** 2024-10-31

**Authors:** Kateryna Metersky, Kaveenaa Chandrasekaran, Yoland El-hajj, Suzanne Fredericks, Priyanka Vijay Sonar

**Affiliations:** Daphne Cockwell School of Nursing, Toronto Metropolitan University, 288 Church St., Toronto, ON M5B 1Z5, Canada; kchandrasekaran@torontomu.ca (K.C.); yolandje@outlook.com (Y.E.-h.); sfrederi@torontomu.ca (S.F.); dr.priyankavso@gmail.com (P.V.S.)

**Keywords:** pregnancy, arrhythmias, ventricular tachycardia, life experience, narrative inquiry, maternal health

## Abstract

**Background:** Approximately half of all women develop palpitations during pregnancy, with a quarter experiencing arrhythmias. While most presentations are benign, some cases can result in sudden cardiac death or serious symptom development. Considering such clinical presentation, healthcare providers must acquire knowledge in this area to provide comprehensive prenatal, perinatal, and postnatal care. However, no study could be located that focused on women’s life experiences of such complications during or in the post-pregnancy period. **Objectives:** The study aims to share the results of a study that explored the life experience of one woman who developed non-sustained ventricular tachycardia during her third pregnancy that lasted into the postpartum period. **Methods:** Using narrative inquiry self-study methodology, a woman’s experiences were explored to uncover the challenges she faced in coping with such complications during a period of transition for herself and her family. This methodology allowed for an in-depth understanding of how these complications could affect all aspects of her life. **Results:** Four narrative threads were produced: (1) diagnostic challenges and delayed recognition; (2) impact on maternal identity and family dynamics; (3) navigating healthcare systems and treatment decisions; and (4) long-term adaptation and resilience. **Conclusions:** The intention was to add to this topic area to ensure future researchers, current and future healthcare providers, and patients have literature they can refer to when studying, providing care for, or experiencing similar health complications. Acquiring this knowledge can aid healthcare professionals to ensure appropriate care is provided, risks are minimized, and their recovery is well supported.

## 1. Introduction

Pregnancy is a life-changing experience affecting women on a physical and emotional level. Globally, 140 million babies are born every year. In Canada alone, the birth rate was 10.072 births per 1000 people in 2023 [1]. The World Health Organization (WHO) (2019) [2] defines maternal health as the health of women during pregnancy, childbirth, and the postnatal period. The postpartum period, also known as the “fourth trimester”, constitutes the phase following childbirth where the physiological changes associated with pregnancy within the woman revert to the nonpregnant state [3].

With the transformative experience of pregnancy comes potential complications that can impact maternal health. In 2020, it was estimated that about 287,000 women died during and following pregnancy and childbirth in the world [2]. In particular, electrophysiological complications can pose a significant risk to the health and well-being of pregnant and postpartum women, as these complications can range from mild disturbances, such as palpitations, to more severe conditions like ventricular tachycardia and ventricular fibrillation [4]. Typically, these complications persist for short periods of time and resolve by the time of delivery [5]. However, at times these complications continue after the end of the postpartum period and can become a chronic health condition. Moreover, although most arrhythmias during pregnancy are common and generally benign [6], some arrhythmias can be a cause for concern. According to Coad and Frise (2021) [3], tachyarrhythmias during pregnancy are an urgent health concern that requires timely intervention as they can result in hemodynamic instability and subsequent placental hypoperfusion, potentially necessitating immediate cardioversion. Sustained palpitations that may arise from ventricular arrhythmias, such as idiopathic ventricular tachycardia (VT) or pathway-related supraventricular tachycardia (SVT) [6], also necessitate prompt medical intervention to ensure the health and safety of the pregnant woman and developing fetus.

Palpitations and dizziness are prevalent symptoms during pregnancy, necessitating vigilant monitoring. A study involving 110 pregnancies with these symptoms and 52 asymptomatic pregnancies revealed that 23% of the control group experienced palpitations within a 24-h Holter monitoring period [7]. The primary rhythm abnormalities noted were premature ventricular or atrial ectopic beats, present in 50–60% of symptomatic pregnant patients, which typically resolve during the postpartum period [7]. Additionally, Joglar et al. (2023) [8] indicated that supraventricular arrhythmias occur in 31–59 per 100,000 pregnancies, with an increasing trend over recent decades. Two percent of pregnant women have also been noted to experience paroxysmal supraventricular tachycardia (PSVT), which is a less common but significant form of sustained tachyarrhythmia [9]. While most pregnancy arrhythmias are benign, Vernekar et al. (2014) [10] reported that some can result in serious symptoms and, in rare cases, even sudden cardiac death.

Considering the prevalence of electrophysiological complications during pregnancy and the postpartum period, it is critical to understand the life experiences of women who continue to live with such complications [11,12]. Acquiring this knowledge can aid healthcare providers to ensure appropriate care is provided, risks are minimized, and patient recovery is well supported. 

### Background

Electrophysiological complications encompass problems that arise within the heart’s electrical system [13]. There are several types of arrhythmias: ventricular arrhythmias, including ventricular fibrillation and tachycardia, and supraventricular tachycardia, to provide some examples. Women are likely to face any of these types of arrhythmias due to the substantial elevations in estrogen and β-human chorionic gonadotropin, while hemodynamic alterations include an increase in circulating blood volume and cardiac output that occur during pregnancy [4]. It is vital for women experiencing any form of arrhythmia to acquire immediate attention and appropriate follow-up, as such conditions can cause hemodynamic instability and subsequent placental hypoperfusion, along with perturbations in the hemodynamic, autonomic, and hormonal systems [3,14]. Without prompt restoration of a normal heart rhythm, these complications can lead to fatal consequences in the matter of minutes [14].

From a scientific perspective, the correlation between the incidence of electrophysiological complications during pregnancy and the postpartum period is evident. However, there is little to no research on the life experiences of women in this predicament. Pregnancy and the postpartum can be a challenging period for women to experience symptoms that already naturally occur (i.e., nausea/vomiting, fatigue, heartburn). Experiencing additional symptoms due to electrophysiological complications can make this period more challenging.

Upon exploration of the literature on the emergence of arrhythmias in pregnant or postpartum women with no history of previous illness, only a handful of studies could be located, which were also included in a scoping review (hidden for blind review) on this topic. Thirteen articles are included in the review, which was conducted using the Joanna Briggs Institute methodology and PRISMA-ScR framework. Ten studies were quantitative in design, while three were discussion papers. The findings of this review revealed various tachyarrhythmias reported among pregnant women or women in the immediate postpartum period, including the onset of supraventricular tachycardia (SVT). Moreover, the literature attributed an increased cardiac output, blood volume, and elevated levels of hormones (estrogen and β-human chorionic gonadotropin) that occur during pregnancy to the incidence of cardiac arrhythmias. Finally, mixed findings were reported on advanced maternal age as a risk factor to develop cardiac arrhythmias during pregnancy. However, due to the lack of qualitative evidence, little is known about the life experiences of a diagnosis of arrhythmias during pregnancy or the postpartum period.

In this narrative inquiry self-study, the experience of one previously healthy woman, with no significant past medical history, who encountered electrophysiological complications during pregnancy and the postpartum period, was explored in great detail. The objective was to elucidate her life experience and highlight the challenges she faced in coping with such complications during a period of transition for herself and her family unit. The intent of this research was to expand this area of research to ensure future researchers, healthcare providers, and patients have a piece of literature they can refer to when studying, providing care for, or experiencing similar health complications.

## 2. Methodology

The qualitative methodology of narrative inquiry [15], with a specific focus on self-study, was employed by the researcher (participant). This approach involves systematically analyzing personal experiences of the world and creating meaning through the study of people’s daily lives as stories. During the exploration of experience, the researcher and participant establish a collaborative relationship; initially, the researcher invites the participant to share their stories about a particular phenomenon of interest. This is followed by both the researcher and participant engaging to develop and construct the importance of the story through collected data and ongoing consultation with each other [16].

This methodology was selected as it is the most effective way of characterizing the phenomena of human experiences, utilizing stories as the primary source of data [17]. Storytelling in educational research is rooted in the idea that humans are inherently inclined towards narratives, both as individuals and as social beings, shaping their life experiences. Consequently, the examination of storytelling entails exploring the various social and cultural ways in which humans perceive and engage with the world [15]. It is important to note that narrative inquiry was selected over other qualitative methodologies as it prioritizes storytelling, ultimately encouraging the participant to share their unique experiences in a narrative format. As opposed to a structured qualitative interview that may focus on specific questions or themes, this method provides participants greater agency in shaping their narratives [16]. In this study, the context was a previously healthy woman who developed electrophysiological complications during pregnancy and the postpartum period. This methodology allowed for an in-depth understanding of how these complications could affect all aspects (i.e., social, emotional, physical, psychological, and spiritual) of the woman’s life.

### 2.1. Sampling and Participant

While this study did have formal inclusion criteria, as the researcher was also a participant in this self-study, it was not used to recruit further participants. The inclusion criteria required for the one study participant to have English language skills and provide consent for participation. 

### 2.2. Setting

The researcher (participant) chose to focus their experience of their electrophysiological complications during the postpartum period after having their third child, which took place in Ontario, Canada, their province of residence. The reflection on experience and this self-study took place online in Ontario, Canada.

### 2.3. Ethical Considerations

This study received Research Ethics Board approval (REB 2023-221) from the principal investigator’s academic institution. Informed consent was obtained from the study participant by the project principal investigator prior to engaging in data collection. The participant was instructed to select a pseudonym to participate in the study to maintain their anonymity and confidentiality. The participant also retained the rights to their creative piece and kept the original version. Only copies were retained by the research team. 

### 2.4. Data Collection

A research assistant and member of the research team conducted the data collection. In a self-study, interviewing is a commonly used method of data collection, whether that is a researcher interviewing themselves or others. The research assistant was selected to conduct the data collection to ensure someone was able to record events, participate in detailed note-taking, carefully analyze the interview, and review the interpretation by the participant (researcher) [18].

The participant was asked to undergo a two-phase data collection process: a semi-structured individual interview immediately followed by the narrative reflective process of self-artistic reflection. During phase one, a semi-structured individual interview was conducted and audio-taped for the purpose of transcription. The interview was approximately 45 min in length. Breaks were given at the participant’s convenience, and the participant was made aware of their ability to withdraw from the interview at any time. The participant was asked a series of questions in relation to the electrophysiological complications that occurred during pregnancy and the postpartum period. The participant was encouraged to share stories, opinions, and personal experiences. During the interview, the participant was asked to refrain from sharing any identifiable information either about themselves or others.

During phase two, immediately after the interview, the participant was asked to engage in an aspect of the narrative reflective process. The researcher invited the participant to create an artistic piece to describe their personal electrophysiological experience (hidden for blind review). The participant was encouraged to create anything that represented their raw emotions and feelings living with this complication.

### 2.5. Data Analysis

When both phases were completed, the research team transcribed the audio recording. Any emotional responses were also included in the transcription, with any confidential or identifiable details omitted. Each research team member made sure to re-read the transcript at least two times to ensure transcription accuracy.

Three types of justification by Clandinin et al. (2007) [16] were used to aid in the analysis process, ranging from a small (individual) to large (societal) scale: personal, practical, and social. Regarding the personal justification, the research team members all engaged in a personal reflection on their emotive and cognitive responses to interacting with the data, which included stories and creative artistic pieces. This process allowed the researchers to identify the meaning of the data in relation to their own personal contexts and experiences, particularly being females of varied reproductive ages and pregnancy experiences. The second type of justification is practical, which requires the researchers to consider how the collected data impact their profession, which in this context is the healthcare sector [16]. In this level of analysis, the research team looked at the data from a broader perspective to answer the question, “What does the collected data mean/add for the healthcare providers and their practice when working with women who are experiencing such complications in the peri and postpartum periods?” In the last level of analysis, the social researchers explored the significance of the findings for the larger healthcare system and societal value, attempting to answer the “So what?” question ([16], p. 25). Here, the researchers were looking at the narrative threads that emerged not only within the participant’s stories to present implications on a larger scale for society as well as nursing and health professions education, practice, policy, and research. Rigor in this study was ensured using a well-documented audit trail to keep a record of the decision-making process throughout the study, including the data collection and analysis processes [19].

Narrative inquiry and self-study studies are often written in the first person [18]. However, this study is written in third person to improve the credibility of the research findings and conclusion. Since the researcher is also the participant in this study, by adopting a third-person perspective, the researcher remains distanced, and it mitigates potential biases by presenting the reflections in a more objective manner [18]. Moreover, the positionality of the researcher (participant) was considered and imperative in implementing this narrative inquiry self-study. As the researcher is also the participant in this study, it is important to recognize that their background, individual health experiences, biases, and assumptions will influence the research findings and conclusions. Positionality was achieved in this study as the researcher (participant) critically reflected and acknowledged any preconceived notions or biases that can shape all aspects of the research process. This includes their position in society, experiences of privilege and marginalization when accessing the healthcare system, and access to services/resources.

## 3. Results

The findings of this qualitative methodology self-study using narrative inquiry [16] include the retelling of the experiences of the researcher (participant) involving their challenges with electrophysiological complications during the postpartum period. The findings are presented in the following order, in line with the narrative inquiry method: story, metaphor, and drawing. During the first part, the story began near the end of the participant’s third pregnancy and ended at present day. In the subsequent part, the participant gave a metaphor for their experiences and a drawing to visualize their experience. The pseudonym of the participant was ‘Matilda’.

### 3.1. Part One

Matilda is 34, and their electrophysiological complication is non-sustained ventricular tachycardia (NSVT). According to Matilda, NSVT, in simple terms, “*… it just means that my heart can speed up without any warning and then eventually it feels like it stops and restarts itself*”.

Matilda’s understanding of what it means to be healthy entailed “*being able to function, play, learn, and contribute to society to the best of my capacity without limitations*”. Things she considered healthy included wearing glasses, managing a moderate amount of stress at her job, eating healthy, and getting exercise on the weekends. From her perspective, things start to become unhealthy when they start to get out of control, for example, not being able to eat nutritious food as a result of being swamped with work. According to Matilda, “*I don’t necessarily look at health as the absence of disease; I also consider everything else that’s going on around me, that I’m getting enough mental stimulation, that I’m able to get fresh air, nutritious foods, … (that I get) time with my family, etc.*”

To Matilda, being pregnant represented a happy period of time because it meant starting or expanding her family and not a time when she would be thinking about complications. Matilda elaborated on this saying: “*I think of Hollywood movies where I have cravings of pickles and ice cream and everything’s rosy, will turn out fine. I’m pregnant for 40 weeks and I’m going to have a good and healthy delivery. As I get closer to that delivery, fear steps in on how that delivery will look like, but generally thinking of pregnancy, I think of everything positive*”.

Matilda first began to experience electrophysiological complications at around 32 weeks of her third pregnancy. She began to have a rapid heartbeat and fainting episodes, the latter of which she defined as zoning out and looking like she had completely dissociated. She described the feeling as “*if a tap is turned on at my toes and all the blood is kind of rushed out through the tap and then when my heart restarts, I feel all the blood rush back to my face*”. After notifying her OBGYN of these experiences, they were not too concerned as many changes would be happening to her body at that time. After asking whether having her initial two normal pregnancies made a difference in determining whether the complications she experienced during her latest pregnancy were abnormal, Matilda said that coming from a nursing background made her disregard the severity of her symptoms. Since she had been familiar with all the physiological changes women experience during pregnancy (and had even worked in labor and delivery prior), she would not have been able to easily find a distinction.

However, one day while she was breastfeeding her newborn, Matilda had a severe fainting spell that resulted in her falling back in her rocking chair with her baby. After this occurrence, her husband insisted on seeking medical attention again to see what the problem was. She first went to a walk-in clinic as it was hard to obtain an appointment at her family doctor at the time. After going through some tests at the walk-in clinic, she was sent to a cardiologist for further care. Matilda received exceptional care at the cardiologist and did all the required tests at the clinic, including an ECG that was unremarkable and blood work that was within normal limits. She was then told to wear a Holter monitor for a two-week period instead of for 48 h, as she was not experiencing frequent fainting episodes. 

While wearing the Holter monitor for two weeks initially, Matilda was not feeling anything unusual. She was excited that the condition was perhaps gone but nervous that the monitor would not have picked up any abnormalities. On the thirteenth day of wearing the monitor, while having dinner with her family, she felt her symptoms again when she went to sit down. She experienced headiness, palpitations, and heart racing against her chest wall. “*At this point, it’s not comfortable because my kids and husband visually saw this. A few days later, the cardiologist calls and says* “*I have good news and bad news, good news we picked something up, bad news we picked something up*”. She was then referred to an electrophysiologist who specializes in postpartum women, etc.

There, the electrophysiologist confirmed the diagnosis, ran a series of tests, put her on several limitations, and gave her medications. She also had to wear a Holter monitor on and off for a span of two years, ranging from a couple hours to a couple months at a time, when her medications were not working or when she felt different symptoms. According to Matilda, what made the difference between having the monitor for two weeks and a month was the availability of the product. “*The hospital has a limited number of Holter monitors, so if they could not accommodate because most patients are on a 48-h rotating window, they already pre-scheduled somebody else, they couldn’t give it to me for that extended period of time*”.

Regarding the limited supply of Holter monitors, Matilda said “… when thinking about that as a human being, that’s scary because I’m a mom with three kids, and thinking that my own health and diagnosis is dependent on availability (of Holter monitors) makes it a bit sad and challenging.” In terms of her experience with the Hotler monitor itself, Matilda experienced challenges, particularly with the Holter stickers/electrodes attaching the monitor wires to her skin. “I have very sensitive skin. I have a condition called seborrhea (which) is more predominant on the face and sometimes the scalp, … but because of that skin condition I’m allergic to the adhesive on the Holter monitor stickers. So, they do have hypoallergenic flares that you can put on the Holter, but although that helps, their adhesive is either not as strong or attaches in a way that starts ripping my skin”. According to Matilda, taking showers was also a challenge, as she had to replace the electrodes every time she took a shower, which was daily in her case. As a result of unsticking and re-sticking the electrodes, she would bleed at the location of the electrode, and her skin would peel. For the Holter monitor itself, she said, “it’s been 2 years, (I still) have scars on my body where you can tell I had the Holter monitors on, embarrassment because of the bulkiness of the equipment and so sometimes (the technician) would put it on a pouch, sometimes a little purse, sometimes they would put it on a belt clip”.

Although she did not wear it during her pregnancy, Matilda claimed that it was not any easier to wear the Holter monitor. “I had three kids at the time I was wearing it, I had a three-year-old, a two-year old and an infant. So, I had to pick them up, carry them and I had to breastfeed the infant who loved the wires. So, one wire comes off and you lose the signal, so there were instances where I had my episode, I pressed the button but one of the wires wasn’t attached. I would wake up in the morning and two of the wires would be off. So, it was really challenging to wear, not just the stigma around it too, a 34-year-old walking around with various wires”.

Matilda described how she felt after being diagnosed with her condition as “Devastated. Nervous. Fear of the unknown. Leaving my husband without a partner to raise three kids… I remember my husband took the kids to get books at the library and I was sitting in the car and I pulled out my cell phone and I recorded a goodbye video to my kids because I was told that I wasn’t going to live long and the rate of dying is very high with this condition. I literally tried everything that I was told to do … (though I don’t do them often) I avoided heat such as saunas and jacuzzis, no coffee, no tea, I completely transitioned into herbal products, and that was hard for me because I really love tea. I gave up meat, lost a lot of weight (because) they thought my weight gain through pregnancy was having complications on my heart”.

She mentioned that she had seen the same video she recorded a few days prior to our interview and said how her life was in a completely different place now, three years later. “*I don’t have the same fears, but at the time of diagnosis, to think I have three boys and to leave them without a mom, like that was terrifying. I wasn’t thinking about losing myself and losing my life, I was thinking about dying and leaving them without a mom. That’s what was hard*”.

In terms of how the diagnosis impacted her family, Matilda wanted to have more children but was concerned about the potential risk it would have toward her health. She described how it was challenging to navigate her emotions and thoughts, as well as support her partner when engaging in all the medical appointments she had not planned for. She had to spend time away from her sons while she was hospitalized or undergoing various procedures, but made an effort to stay with her infant to breastfeed. Since Matilda preferred to breastfeed instead of using formula, this was also impacted as she was required to take a dye for her MRI. “*The dye can be damaging to the infant and it passes through breast milk, so I remember feeling so guilty about having to give him formula for that day. And what if he gets addicted to formula and won’t want to be breastfeed anymore. And he was screaming because he didn’t like the formula, he didn’t like the bottle, he wasn’t used to a bottle, I don’t pump either, it was strictly breast. So, I remember that was a very challenging period of “how do I meet my own health needs, at the same time how do I meet the needs of my infant and my family?*”.

Regarding how her diagnosis impacted her relationship with her partner, Matilda said, “He was very supportive, I think he was very terrified and scared himself with the thought of losing me. The good thing is he stayed with the older two boys and I could be focusing on the infant. He met my needs as a mother, bringing me the boys from time to time when it was appropriate. At home, he took a larger load on things because there would be periods where I was too tired—why I was feeling like that was because at the time of diagnosis they put me on bisoprolol, a medication to help with the symptoms but that medication made me very tired”. Her medication made her easily short of breath and resulted in her having very low blood pressure. She did not have a typical reaction to the medication, as even the smallest dose had a great impact on her body. “So there would be periods where after a few hours of being with the kids, I would have to lie down. On the weekends I couldn’t always be left alone with three kids, my husband was working a lot, studying on the weekends so it was hard on our family dynamic but he really stepped up as a partner. If I didn’t have such a supportive partner, it would have been very hard, like somebody, like a single mom to go through this alone”.

Matilda recounted an episode in Fall 2021, where she was at work in her office with no one around, as the COVID-19 pandemic was rampant at the time. She had a colleague check on her after the episode, who walked with her to the subway. Right as she was about to get on the subway, she decided to go to the hospital because of the severity of her episode. After doing all the necessary tests, she was told that she needed to have a loop recorder put into her body, which Matilda described as a “*Holter monitor for two years*”. 

The loop recorder, which is connected to her cell phone, has to be removed every two years to replace the battery. Since having the loop recorder, she has only pressed the button to activate the device a handful of times. According to Matilda, her condition has gotten significantly better to the point she rarely has the sensation of her “*heart whacking against her chest wall*”. After pressing the button, the loop recorder application would tell her that she has tachycardia instead of non-sustained ventricular tachycardia since her heart would return to a normal rhythm instead of “restarting”. To explain this, Matilda has a theory: “*I don’t know (if this) theory is right, but it’s because I had the three kids so close in a four-year period and the pregnancy hormone, oxytocin, I have a feeling (is what) was the cause of this. I got the loop recorder implanted as I was finishing breastfeeding and since that time because I finished breastfeeding, I’m not pregnant—my body is not releasing the same high levels of oxytocin anymore. So I feel like the longer I went since having given birth, the better I started to feel*”. Matilda now has a biannual follow-up with an electrophysiologist to redo her tests, but is optimistic. “*I’ll know next summer when I go for various tests again but I feel much more confident that I can live a very long life. And that’s a feeling, I don’t know if I feel that because I’m healthy, because I’m taking care of myself*”.

Regarding Matilda’s message to mothers who may have similar symptoms or undergone similar complications, Matilda says not to wait. “*The minute you feel something is wrong, go seek help because it could be dangerous to you, it could be dangerous to your child. I didn’t think much of it when I was pregnant but what if I fainted? Fell on my stomach? Hurt my offspring in any way? So, seek help the first time it happens. If you feel like it’s not something you should be experiencing, go seek help*”.

Matilda also said to stand your ground and advocate for the best possible treatment you can obtain. “*You know your body best, for me medication did not work. I had to go on and do something else, use different treatment modalities to ensure I was okay. Everybody is different, everybody is an individual. So, listen to your body, do what’s right for you and your family and at the end, in collaboration with your medical team and the advice of the specialists of course*”.

The third piece of advice is to ask for information. “*Get all your answers right at the beginning because for me, to this day I don’t really understand that condition… I had to ask a gazillion questions to my electrophysiologist and I always felt like I was an inconvenience… I wasn’t given a resource package or pamphlet and it makes it very scary because you rely on Google. Thankfully I’m a nurse so I had access to credible sources, but if you are not, you go to talk to people, you go on Google, you go on Reddit, you read all this health misinformation that can add so much fear. I was really scared of what living with this condition would mean for my family and myself and my longevity, so I would really suggest pushing for that information*.”

### 3.2. Part Two

For the second part of the interview, Matilda chose to describe her experience through a metaphor. “*I have this plant in my house that my father got for me a while ago before I even had kids. I have a love for palm tree plants but because my apartment was very small my dad got me a little palm tree plant in a pot and (when) I had one kid and it was blooming nicely. Second kid, it was blooming nicely, but during my third pregnancy, it started to wilt a little bit and get yellow. I think it’s because I just didn’t have time to water it and take care of it the more kids I had. So, for you today, the metaphor I have is that plant. It was green; it continues to grow and continues to add more branches and leaves, but if it’s not taken care of, if it’s not feeling well, it starts to turn yellow. You can still salvage it even though once it turns yellow, you’re really going against the very tall mountain to salvage it, but you still can before it completely starts falling apart*”.

Matilda further goes on to describe the direct parallel between her health and the health of the palm tree. “*The leaves start falling apart, and the stems completely become very yellow or dark brown and it’s impossible to save at that point. That’s my metaphor of my experiences. I was very healthy, always took care of myself, never thought that I could turn yellow. I thought I was like that beautiful plant that I always wanted, pursuing things in life that I always wanted, getting married, having children, getting the career that I am very proud of but I turned yellow and I got this condition*”.

Recently, Matilda has said how the plant has started to become green again, in line with how her life is now. “*I’m starting to see that plant is turning green, there are fresh stems that are growing out of it now and so I feel like yes there is still evidence of that yellow but there’s a new beginning, new stems are coming out, new stems are growing*”. While the plant has become green, there are still yellow leaves on the plant, which Matilda sees as a reminder to water and continue to support her plant, along with her own health and wellbeing. “*I wanted that memory to be there to understand the difficult part of that plant’s life but at the same time, take beauty in the growth and the new things that are coming out in the possibilities there, so that’s the same with me for the rest of my life. Whether I want to or not I’m gonna have a scar on my chest on the left-hand side where they inserted the loop recorder and I get discriminated against all the time*”.

Matilda describes the challenges she still faces with her loop recorder. “*Whenever I travel through airports they’ll look at it and think I’m hiding something… I just came back from a trip to Qatar and I was traveling through Germany and I explained to them I don’t feel comfortable going through metal detectors, I would prefer a pat down, they actually called the police on me. They put me in a room and asked me to take off all of my clothes. I stood there, naked with two police officers with guns. One of them, poking their fingers into my chest to feel the battery to be convinced that I have a battery implanted versus just looking at my card that says I have an implantable device, so I get continuously reminded, and even when that implantable device is gonna be removed from my body I’m gonna have that scar of where it was*”.

Despite the challenges she may face, Matilda continues to maintain hope for herself and her palm tree. “*I will continue to hopefully live a very healthy life and so those green leaves are me coming out of this very unfortunate situation but the yellow is that scar that I will always have. I will get to see my kids grow up. Hopefully I will be there for them and hopefully those green leaves continue to grow, and I will continue to evolve in various ways that I meant to*”.

## 4. Discussion

This narrative inquiry self-study and exploration of the storied life experience of one participant who experienced electrophysiological complications during pregnancy and the postpartum period produced four key narrative threads: (1) diagnostic challenges and delayed recognition; (2) impact on maternal identity and family dynamics; (3) navigating healthcare systems and treatment decisions; and (4) long-term adaptation and resilience. 

### 4.1. Narrative Thread #1: Diagnostic Challenges and Delayed Recognition

One prominent narrative thread in Matilda’s story was the initial difficulty she experienced in recognizing and diagnosing her cardiac symptoms. Despite experiencing rapid heartbeat and fainting episodes starting at 32 weeks of pregnancy, her concerns were initially dismissed as normal pregnancy-related changes. This aligns with the existing literature showing that cardiovascular symptoms in pregnant and postpartum women are often overlooked or misattributed to common clinical manifestations of pregnancy [20]. Matilda’s background as a nurse may have further complicated the diagnostic process, as her familiarity with normal physiological changes in pregnancy led her to downplay her symptoms. The delay in diagnosis highlights the need for increased awareness among healthcare providers about the potential for cardiac complications in pregnancy and the postpartum period [21,22]. Matilda had to unfortunately experience a fainting episode while breastfeeding her newborn. This incident raises the risk of how dangerous undiagnosed cardiac conditions can be, not only for the woman experiencing them but for their fetus or newborn. Recent guidelines emphasize the importance of thorough cardiovascular assessment in pregnant women presenting with symptoms such as palpitations or syncope [23]. Matilda’s experience underscores the value of listening to women’s concerns and investigating persistent symptoms, even in the absence of traditional risk factors. The turning point in Matilda’s diagnostic journey came postpartum, when she fainted severely while breastfeeding her newborn, after which her husband insisted that she seek further re-examination. Her experience demonstrates the necessity for awareness of the risks of undiagnosed cardiac conditions immediately postpartum, not only to a mother but also to the fetus/newborn [10]. Through the application of the narrative inquiry method, the detailed account of one participant’s life experience reveals critical concerns, such as the delays in diagnosing and treating cardiac arrhythmias during pregnancy, which resonate with existing evidence in the literature [24]. 

### 4.2. Narrative Thread #2: Impact on Maternal Identity and Family Dynamics

Another significant thread in Matilda’s narrative was the profound impact of her diagnosis on her sense of self as a mother and the relationships within the family unit. The sudden transition from a joyous expectation of expanding her family to facing a life-threatening condition created significant emotional turmoil in Matilda. Her story of experience is riddled with examples of actions Matilda had to perform that went against the conventional actions of a mother in the postpartum period. Instead of taking the time to focus on her newborn and the recovery her body needed, Matilda spent her time attending medical appointments, experiencing hospitalizations, having to pause breastfeeding to undergo diagnostic tests, and experiencing prominent lifestyle modifications. This points to a challenge Matilda experienced with balancing her own health needs with those of her infant, particularly in relation to breastfeeding. Her distress at having to temporarily switch to formula due to needing to take a contrast agent for an MRI procedure highlights the emotional complexity of infant feeding decisions for mothers with health complications. This aligns with research showing that interruptions to breastfeeding due to maternal illness can be a significant source of distress and guilt for mothers in the postpartum period [25]. Specifically, no research could be located that explored the impact that maternal electrophysiological complications during the pregnancy and/or postpartum period can have on maternal and familial mental health and wellbeing, bonding with the newborn, and transition to an expanded family unit.

Matilda’s description of recording a goodbye video for her children vividly illustrates the fear and uncertainty she experienced. This finding aligns with research on the psychological impact of pregnancy complications, which has shown that women often experience a sense of loss, guilt, and anxiety about their ability to fulfill the maternal role [26]. Particularly, her condition brought on symptoms such as shortness of breath as a result of a medication side effect that placed limits on her activity level and involvement with childcare, requiring changes to existing family member roles. This change in roles can further have implications on Matilda’s emotional and mental wellbeing. A study exploring role changes in women by Harrington et al. (2022) [27] investigated how women’s self-esteem was impacted when they experienced gender role discrepancy when they were unable to fulfill traditional feminine roles. The study findings indicated that such a discrepancy negatively impacted their self-esteem and caused a strain on their relationships within their families [27]. 

Not only that, but this role change placed extra responsibilities on Matilda’s partner. Research on the impact of increased caregiving responsibilities on fathers when their partner is ill has found that such roles significantly exacerbate fathers’ psychological distress and physical health issues [28,29]. Fathers experience heightened levels of stress, anxiety, and depression, along with increased physical health problems due to the dual burden of caregiving and household responsibilities. Additionally, these challenges can lead to strained family dynamics and reduced overall well-being of the entire family unit [28,29]. 

Finally, it is vital to also consider the impact of a change to the maternal role on the children within the family unit. Matilda already has two other children who were old enough at the onset of the symptoms to recognize that something was wrong physically with their mother. A study by Horner (2019) [30] found that maternal chronic illness significantly affects children, with children of chronically ill mothers being at an increasing risk for emotional and behavioral issues, as well as language and cognitive difficulties. Corresponding to these findings, a study by Mudiyanselage (2024) [31] highlighted that children of mothers with chronic illness often experience heightened psychological distress and developmental delays. These studies collectively underscore the substantial adverse impact of maternal illness on children’s emotional and cognitive well-being. Matilda’s experience highlights the far-reaching impact of maternal health complications on the entire family unit, affecting her maternal role identity, the partner role, and her children’s well-being. This underscores the critical need for healthcare providers to adopt a holistic approach, offering comprehensive support not just to the affected mother but to the entire family, addressing both physical and psychological needs throughout the challenging journey of maternal illness and recovery. Narrative inquiry captures complex phenomena that include emotions, ultimately providing insight into the psychological distress that has occurred throughout the life experience of the participant during this pregnancy that reflects broader trends in previous research. 

### 4.3. Narrative Thread #3: Navigating Healthcare Systems and Treatment Decisions

Matilda’s experience navigating the healthcare system and making treatment decisions forms another important narrative thread. Her story reveals the challenges of coordinating care across multiple specialists and managing medication side effects. The difficulties she encountered in obtaining extended Holter monitoring due to limited equipment availability highlight systemic issues in healthcare delivery for women with pregnancy-related cardiac complications. Matilda’s eventual transition to an implantable loop recorder illustrates the potential benefits of newer technologies in managing cardiac arrhythmias. However, her experience also underscores the need for comprehensive patient education and support when introducing such devices. Matilda’s description of feeling like an “inconvenience” when asking questions about her condition points to gaps in patient-centered care and communication. These findings align with research showing that women with pregnancy-related health complications often feel overwhelmed by the healthcare system and struggle to obtain clear information about their condition and treatment options [6]. Matilda’s advice to other women to “stand your ground and advocate for the best possible treatment” reflects the importance of patient empowerment and shared decision-making in perinatal care. It has been found that engaging patients in their care significantly improves health outcomes, increases treatment adherence, and enhances patient satisfaction [32,33]. Patients who participate in shared decision-making processes are more knowledgeable about their conditions and treatments, leading to more realistic expectations and reduced anxiety. Additionally, involving patients in their care decisions results in higher adherence with treatment plans and a lower risk of complications [32,33]. The participant’s narratives, alongside insights from the relevant literature, underscore the vital role of patient education in empowering pregnant women diagnosed with cardiac arrhythmias to effectively navigate healthcare systems and make informed decisions about their care. 

### 4.4. Narrative Thread #4: Long-Term Adaptation and Resilience

The final narrative thread that emerges from Matilda’s story is one of long-term adaptation and resilience. Although Matilda continues to have such symptoms, the symptoms are not as severe as when she was breastfeeding. Matilda also visits the electrophysiologist every three years and can schedule an appointment whenever needed if the symptoms reoccur in the same magnitude. Matilda usually experiences symptoms of NSVT only during vigorous exercise or when she is under increased stress at work. Her description of gradually improving symptoms and increasing confidence in her longevity illustrates the potential for positive outcomes even in the face of serious perinatal or postpartum health complications. This aligns with research on post-traumatic growth in women who have experienced pregnancy complications, which has shown that women can develop increased personal strength and appreciation for life through the process of overcoming health challenges [34]. The palm tree metaphor Matilda had discussed to draw similarities between the plant and its growth and development with her journey of experiencing maternal health vulnerability and the potential for restoration underscored that if correctly cared for and nourished correctly, both would lead to a successful long-term outcome. This was akin to Matilda coming from the perspective of being resilient among the health challenges she was experiencing to demonstrate personal growth during this difficult period. In addition, Matilda’s theory about the potential role of pregnancy hormones in her condition and the fact that she had three of her children so close together, while speculative, raises questions for future research to address the complex interplay between reproductive physiology and cardiac function. 

Matilda’s experience highlights the need for long-term follow-up pregnancy-associated cardiac arrhythmias to ensure successful adaptation to this diagnosis occurs, and Matilda is able to ‘live well’ with this condition in the community. Living well with a new chronic condition diagnosis is challenging, requiring acceptance, coping, self-management, integration, and adjustment [5]. The key piece about living well with a chronic condition is recognizing that this process is cyclical, not linear, in nature and that as life advances with this condition there will be ebbs and flows of Matilda needing to accept, cope, self-manage, integrate, and adjust [5]. A prime example of this is the presence of the condition at the forefront. Every three years, Matilda needs to meet with her electrophysiologist to do a series of tests and receive a check-up. As well, each symptom brings the condition to the forefront and requires Matilda to re-evaluate her level of adaptation and resilience to continue to live well with the chronic condition. 

Each patient or person will present their own unique level of difficulty in managing Ambrosio et al.’s (2015) [5] attributes. Some may achieve a sense of ‘The New Normal’, where they successfully integrate the condition into their lives, while others might struggle, leading to states of disavowal, false normality, or disruption. This process demands resilience, support, and continuous effort from the patients and caregivers, making it a significant challenge in their life journey [32]. Ultimately, the participant’s individual narratives align with findings from existing research, illustrating the resilience of pregnant women in the face of significant stressors and life changes during pregnancy and the immediate postpartum period [35]. 

### 4.5. Implications

When considering the findings of this narrative inquiry self-study, significant implications emerged for collaborative practice and care delivery. Considering Matilda’s illness experience and the myriads of healthcare providers that were involved in her care, including obstetricians, primary care physicians, cardiologists, electrophysiologists, and nurses, it is vital that continuity of care and effective communication among providers and between providers and patients as well as their family members occurs. This will not only ensure the delivery of comprehensive care to the patient but also lead to the minimization of gaps in care navigation to be experienced. Following the principles for collaborative practice outlined by the Canadian Interprofessional Health Collaborative’s National Interprofessional Competency Framework is an ideal way to ensure this occurs within an interprofessional team. 

In addition to effective collaboration with the interprofessional team, to facilitate early detection and intervention, healthcare providers must conduct a thorough cardiac evaluation for all pregnant women, even if they do not have any significant cardiac history. Pregnant women must be regularly monitored throughout their pregnancy to detect any changes in cardiac arrhythmias or symptoms. Education must be provided to pregnant women about the symptoms of cardiac arrhythmias, potential risks, and symptoms to ensure they understand the timely significance of contacting their healthcare provider and appropriate follow-up can be provided. 

In terms of policy implications, this study’s findings point to the need for organizations and physicians in specialties that can provide care to women in the postpartum period to expand their policies on the ‘typical’ postpartum check-up period and the potential creation of a care pathway or protocol for care that can be used in clinical practice. In most instances, this follow-up period lasts, on average, six weeks after the delivery of the infant [36,37]. Considering additional follow-up check-points with the mother further into the postpartum period can aid in potential identification of such complications, as the one Matilda experienced in this study [38]. The focus of such checkpoints can be on postpartum care that integrates a cardiovascular risk assessment component. An alternative to this approach can be the development and delivery of programming that focuses on postpartum cardiovascular screening. Innovations in technology can be used to deliver such programming, including the use of telemedicine and home monitoring technologies. The study by Wongvibulsin et al. (2019) [39] demonstrated that connected health technology significantly aids in monitoring cardiovascular conditions by enabling continuous data collection, which facilitates early detection and timely intervention. Moreover, the use of wearable devices and mobile health applications enhances patient engagement and adherence to treatment regimens, thereby improving overall disease management and outcomes [39]. 

Moreover, the findings reveal the vital need for an enhancement to the current education offered in the areas of cardiovascular health, electrophysiology, and maternal-child health. Healthcare providers, including current practicing nurses and nursing students, need proper training regarding the pathophysiological changes occurring during pregnancy and the postpartum period and the implications of such changes on maternal (and fetal/infant) health and wellbeing. Such education should also include how to provide patient teaching on this condition and deliver care that is centered on the person and their family, as this condition impacts the entire family unit. Providing training in how nurses can deliver supportive care to the mother experiencing such complications and providing information for the mother to be able to discern between normal changes occurring during pregnancy and ones warranting further investigation is vital. 

In particular, healthcare providers must be aware of the 2023 Heart Rhythm Society Expert Consensus Statement on the management of cardiac arrhythmias during pregnancy when delivering care to and with pregnant women [4]. It is a comprehensive guide that can be referenced by cardiac electrophysiologists, nurses, cardiologists, and other members of the interprofessional healthcare team on the management of cardiac arrhythmias in pregnant women. It covers important concepts related to cardiac arrhythmias in pregnant women, complications, recommendations for optimal approaches to diagnosis, treatment options (invasive and non-invasive), and patient-specific considerations for treating cardiac arrhythmias in the population of interest [4]. 

### 4.6. Limitations

While the self-study design yields rich and extensive data exploring key elements of a participant’s life experience, it poses limits with the transferability of study findings. The experiences of one individual may not be representative of other women experiencing discomfort as a result of perinatal or postpartum electrophysiological symptoms. Therefore, the conclusions of this study may not be easily generalizable to larger populations or different contexts. Additionally, the participant included in this self-study is a registered nurse who has prior experience working in a labor and delivery care setting. For this reason, this participant may have a more baseline understanding of the experience of pregnancy and potential complications that can influence maternal health compared to the general population. More research is needed to explore the experiences of pregnant women who have developed a cardiac arrhythmia during pregnancy who did not have knowledge of cardiac arrhythmias and pregnancy to elucidate their life experience and highlight the challenges they faced in coping with such complications during a period of transition for themselves and their family unit. Moreover, future research that is focused on obtaining a range of experiences pregnant and postpartum women could have associated with tachycardia is warranted. Women’s perceptions and management of these symptoms may be influenced by cultural background, social level, and pre-existing health issues, among other factors. All of these factors can add to our understanding of the impact such a complication can have on these women’s health, mental, social, and family wellbeing. 

Furthermore, its retrospective design likely introduces recall bias, thus weakening the accuracy of the retold experience. Another limitation relates to the absence of objective physiological data that could not be accessed and evaluated by the research team under the selected methodology used to conduct this study. This type of data would have offered an opportunity to compare and further add to the analysis of the participant’s subjective recall of experience [40,41].

Notwithstanding these limitations, this study provides a useful foundation for future research into the electrophysiological and cardiovascular health of women during pregnancy or the postpartum period. It draws attention to the necessity of a more comprehensive postpartum care strategy that takes the long-term cardiovascular and electrophysiological effects of pregnancy and childbirth into account.

## 5. Conclusions

This exploratory study of one woman’s life experience of electrophysiological complications during pregnancy and in the postpartum period. The process from symptom emergence to diagnosis and subsequent condition management was engulfed with a ‘battle’ between focusing on maternal health and welling and the impact on the family unit. This study underscores the critical importance of early recognition and intervention in pregnancy/postpartum-related electrophysiological complications [42]. The participant’s initial perception of her symptoms, coupled with a nursing background and uncomplicated pregnancies before this experience, exposes a knowledge gap among healthcare providers. This further accentuates the call for an increased level of education and vigilance about atypical cardiovascular symptoms during pregnancy and postpartum periods, not only among healthcare providers but also among expectant mothers.

The study has also shed light on practical difficulties in diagnostic procedures such as skin irritation, lifestyle inconveniences, and social stigmatization due to wearing a Holter monitor, emphasizing the need for more patient-friendly monitoring solutions. Transition to an implantable loop recorder provided more comfort and efficiency of diagnosis, thereby showing a possible way for the development of long-term cardiac monitoring technologies. Notably, a key theme is the profundity of such complications on maternal identity and family life, such as the fear of leaving the children without a mother and the challenges in maintaining breastfeeding during treatment. All these findings call for an integrated approach to maternal care that focuses on physical, psychological, and social health dimensions [43,44]. 

Future research needs to be carried out on larger scales to analyze the incidence, risk factors, and long-term outcomes of pregnancy/postpartum-related electrophysiological complications. Targeted educational programs for healthcare providers and mothers might enable earlier detection and intervention. This study draws attention to the complexity of maternal health and the importance of including women’s voices in the development of healthcare practice and policy. 

## Data Availability

The original contributions presented in the study are included in the article; further inquiries can be directed to the corresponding author.

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
