# Peer review of "Exploring the Life Experiences of Living with Cardiac Arrhythmia Developed During Pregnancy"

_healthcare, 2024, doi:10.3390/healthcare12212178_

Round 1
Reviewer 1 Report
Comments and Suggestions for Authors
This paper is a unique presentation of a case report with an in-depth discussion of a single participant's experience. The following are suggested:
1. Since the participant in this paper is one of the researchers, her baseline knowledge and expectations may not be representative of the general population. Would the researchers be able to include a second participant who is not in the healthcare field? If not, then discuss this as one of the limitations of the case report.
2. The findings on this case report must be compared with published data and provide more details on the new information provided by this unique methodology.
3. Additional clinical history must be provided.
4. Please provide more details what tests were used to confirm the participant's diagnosis.
5. Please provide more details on "long-term adaptation" of the participant and provide some follow up information.
Reviewer 2 Report
Comments and Suggestions for Authors
The study addresses a relevant topic, especially considering that arrhythmias during pregnancy are little studied from the patient's perspective. However, I see several points that could be improved to give more solidity to the work.
1. Methodology: Although the Narrative Inquiry is a good choice to explore personal experiences, the article does not adequately justify why this approach was preferred over other qualitative methodologies. A multiple case study or interviews with more patients would have provided a broader and more robust view. This would be key to better support the findings.
2. Generalizability of the results: Since the findings are based on a single patient, the conclusions are not easily generalizable. It would be advisable to include a more detailed discussion of the limitations and how these affect the applicability of the study in a broader clinical context.
3. Clinical perspective: The study focuses a lot on the emotional experience of the patient, which is important, but a more technical discussion on the medical management of arrhythmias in pregnancy is missing. Including references to current guidelines or previous studies on the clinical management of these complications would give greater weight and relevance for health professionals.
4. Emotional impact and adaptation: Although the emotional impact is mentioned, more could be done on how these experiences affect the patient's life in the long term, both psychologically and in her family environment. This would make the narrative more complete and useful for the reader.
5. Clinical applications: The article mentions that it is intended to be a reference for health professionals, but lacks clear and practical recommendations for clinical care. It would be useful if the manuscript ended with specific suggestions to improve early detection and support for patients in similar situations.
Overall, the article has potential, but needs a greater focus on clinical aspects and a deeper reflection on its limitations. With these adjustments, I believe it can be a valuable contribution to the literature on maternal health and arrhythmias.
Reviewer 3 Report
Comments and Suggestions for Authors
Review comments
The self-study aimed to narrate the life experiences of a woman living with arrhythmia developed during the pregnancy period. To make the work better, I have made the following comments:
Title: Exploring the Lived Experience of Living with Cardiac Arrhythmia Developed During the Pregnancy Period. This can be modified as “Exploring the life Experiences of Living with Cardiac Arrhythmia Developed During the Pregnancy”
Introduction
Line 33: The reference has no date
Line 63: Change lived to life. Apply in other places where this appears
Methods
Okay
Results
Line 207: Reference this “Narrative Inquiry method:”
Line: 212: Change “their” to “her”
The narrative generally well-written
Discussion
Well discussed
